# Transcriptome Analyses of *Diaphorina citri* Midgut Responses to *Candidatus* Liberibacter Asiaticus Infection

**DOI:** 10.3390/insects11030171

**Published:** 2020-03-07

**Authors:** Hai-Zhong Yu, Ning-Yan Li, Xiang-Dong Zeng, Jian-Chun Song, Xiu-Dao Yu, Hua-Nan Su, Ci-Xiang Chen, Long Yi, Zhan-Jun Lu

**Affiliations:** 1College of Life Sciences, Gannan Normal University, Ganzhou 341000, China; yuhaizhong1988@163.com (H.-Z.Y.); 18435203582@163.com (N.-Y.L.); zxdong224@163.com (X.-D.Z.); 15727758058@163.com (J.-C.S.); yuxiudao@163.com (X.-D.Y.); huanansu211@163.com (H.-N.S.); yilongswu@163.com (L.Y.); 2National Navel Orange Engineering Research Center, Gannan Normal University, Ganzhou 341000, China; 3Fruit Bureau of Ganzhou, Ganzhou 341000, China; chencxjx@sina.com

**Keywords:** *Diaphorina citri*, transcriptome sequencing, *C*Las infection

## Abstract

The Asian citrus psyllid (ACP), *Diaphorina citri* Kuwayama (Hemiptera: Liviidae), is an important transmission vector of the citrus greening disease *Candidatus* Liberibacter asiaticus (*C*Las). The *D. citri* midgut exhibits an important tissue barrier against *C*Las infection. However, the molecular mechanism of the midgut response to *C*Las infection has not been comprehensively elucidated. In this study, we identified 778 differentially expressed genes (DEGs) in the midgut upon *C*Las infection, by comparative transcriptome analyses, including 499 upregulated DEGs and 279 downregulated DEGs. Functional annotation analysis showed that these DEGs were associated with ubiquitination, the immune response, the ribosome, endocytosis, the cytoskeleton and insecticide resistance. KEGG enrichment analysis revealed that most of the DEGs were primarily involved in endocytosis and the ribosome. A total of fourteen DEG functions were further validated by reverse transcription quantitative PCR (RT-qPCR). This study will contribute to our understanding of the molecular interaction between *C*Las and *D. citri*.

## 1. Introduction

Huanglongbing (HLB) is a destructive disease of citrus that represents a major threat to the world’s citrus industry. HLB is caused by the phloem-limited, Gram-negative bacterium “*Candidatus* Liberibacter spp” (*C*Las). HLB nearly destroyed the citrus industry in Florida (USA) and most citrus-producing regions of the world [1,2]. *C*Las can decrease plant growth vigor, and ultimately result in the death of the infected citrus tree [3]. There are no effective methods to control HLB once established. Recently, many antibiotics have been adopted to control *C*Las bacteria; examples include the trunk injection of penicillin, streptomycin and oxytetracycline hydrochloride [1,4]. However, these approaches do not work well in the field. The killing of vectors is an effective method to reduce HLB spread [5].

The Asian citrus psyllid (ACP), *Diaphorina citri* Kuwayama, is the principal transmission vector of HLB [6]. Vector control of *D. citri* is recognized as a key approach to preventing the spread of HLB. The application of insecticides is the most widely employed option for reducing *D. citri* populations in some citrus-growing regions [7]. However, the improper use of such chemicals has caused the poisoning of farmers, environmental pollution and insect resistance [8,9]. Some researchers have focused on the olfactory system of *D. citri*, revealing a suite of odorants that can be used to develop affordable and safe odor-based surveillance for *D. citri* control [10]. Yu et al. revealed that the silencing of *D. citri muscle protein 20* (*DcMP20*) resulted in significant mortality and reduced the body weight of *D. citri* [11]. By RNA interference (RNAi) technology, the knockdown of *D. citri tropomyosin1-X1* (*DcTm1-X1*) significantly increased the mortality rate of nymphs [12]. Although some success regarding the control of *D. citri* has been achieved, the threats caused by HLB are going to continue. It is essential to investigate the molecular mechanisms of HLB transmission in *D. citri*. 

Both *D. citri* adults and nymphs can transmit *C*Las. Recent studies suggested that the pathogen multiplies in *D. citri* after acquisition and becomes distributed among various internal tissues, including the alimentary canal, salivary glands, hemolymph, filter chamber, midgut, fat body, muscle tissues and ovaries [13,14,15]. Among these tissues, the midgut is the first barrier that the bacterium must breach before entry into the hemolymph, indicating that the midgut plays an important role in *C*Las infection [16]. Pathogens can manipulate the host’s essential biological processes during the host-pathogen interaction, e.g., by changing protein translation, vesicular transport and protein metabolism [17,18]. In the midgut, many genes and proteins are involved in the defense against pathogens. Wang et al. identified 869 differentially expressed genes (DEGs) in the *Bombyx mori* larval midgut via comparative transcriptome analysis. The results showed that many of the DEGs were associated with protein metabolism, the cytoskeleton and apoptosis [19]. Bao et al. performed a transcriptome-wide analysis on the *Nilaparvata lugens* intestine, and a total of 33 digestion-related genes, 25 immune response genes and 27 detoxification-related genes were identified [20]. The interactions between *D. citri* and *C*Las have become a heavily researched topic regarding the molecular mechanisms of HLB transmission. In recent years, high-throughput omics technologies have been widely used to study the interactions of vector-pathogens. Combining transcriptomics with proteomics methods, Kruse et al. identified many DEGs involved in the TCA cycle, iron metabolism, insecticide resistance and the insect immune system in the *D. citri* gut after exposure to *C*Las [21]. Lu et al. also identified 62 DEPs from uninfected and *C*Las-infected adult *D. citri*, using two-dimensional gel electrophoresis. These DEPs were associated with energy metabolism, host detoxification processes and the cytoskeleton [22]. The hemolymph is a critical component of the *D. citri* immune system, where it coordinates the insect’s immune system activity. Kruse et al. investigated the effect of *C*Las exposure on the *D. citri* hemolymph, using high-resolution quantitative mass spectrometry, and the results revealed that proteins associated with fatty acid synthesis and energy metabolism were upregulated [23]. However, many confusing issues remain to be addressed, such as the mechanism underlying *C*Las entry, multiplication and dissemination in the midgut of *D. citri*. 

In this study, we performed a comparative transcriptome analysis to determine the specific responses of *D. citri* to *C*Las infection in the midgut at the mRNA level. A total of 778 DEGs were identified, and these genes were mainly involved in ubiquitination, the immune response, the ribosome, endocytosis, the cytoskeleton and insecticide resistance. This study established a foundation for further understanding the interaction mechanisms between *D. citri* and *C*Las.

## 2. Materials and Methods 

### 2.1. Insect Rearing, Tissue Collection and Total RNA Extraction

*D. citri* was reared in mesh cages on *Murraya exotica* at 27 ± 1 °C, 70% ± 5% relative humidity, and a 14:10 (light–dark) photocycle. Healthy *D. citri* adults were transferred to *C*Las-infected Newhall Navel oranges and maintained under the same rearing conditions. One hundred *D. citri* from *C*Las-infected Newhall Navel oranges were randomly collected for calculating the infection rate. In general, the head of *D. citri* was cut off using a razor blade, and the genome DNA was extracted using the TIANamp Micro DNA kit (TianGen Biotech, Beijing, China). Afterwards, a PCR reaction was performed to amplify the 16S ribosomal DNA fragment using the HLB pathogen specific primer set OI1/OI2c (forward primer 5′-GCGCGTATGCAATACGAGCGGCA-3′ and reverse primer 5′-GCCTCGCGACTTCGCAACCCAT-3′) based on a previous protocol [24]. Then, agarose gel electrophoresis was performed to investigate the *C*Las infection rate. When the *C*Las-infected percentage reached above 80%, the infected *D. citri* were selected. The uninfected and *C*Las-infected *D. citri* were collected and dissected to obtain the midgut, which was then washed with precooled DEPC-water to remove the remaining tissues. Genomic DNA was isolated from the *D. citri* midgut, and a PCR was performed to confirm the *C*Las infection rate.

For transcriptome sequencing, three groups, each consisting of three hundred *C*Las-infected or uninfected *D. citri*, were dissected for RNA extraction. In order to avoid RNA degradation, the collected midgut samples were stored in RNAlater and frozen at −80 °C. Total RNA was extracted from the midgut of uninfected and *C*Las-infected *D. citri* using an animal tissue total RNA kit (Simgen, Hangzhou, China). RNA concentration and purity were assayed using a NanoDrop 2000 spectrophotometer (Thermo Fisher Scientific, New York, NY, USA) at absorbance ratios of A_260/280_ and A_260/230_. The integrity of total RNA was confirmed using agarose gel electrophoresis.

### 2.2. Library Preparation and Illumina Sequencing

Transcriptome sequencing was conducted on the Illumina HiSeq platform (Novogene Bioinformatics Technology Co., Ltd. Tianjin, China). Sequencing libraries were generated using the NEBNext1Ultra™ RNA Library Prep Kit for Illumina1 (NEB, Ipswich, MA, USA). The library quality was assessed on the Agilent Bioanalyzer 2100 system. The clustering of the index-coded samples was performed on a cBot Cluster Generation System using TruSeq PE Cluster Kitv3-cBot-HS (Illumina, San Diego, CA, USA). After cluster generation, the library preparations were sequenced on an Illumina Hiseq platform, and 125 bp paired-end reads were generated. The sequencing fragments were translated into fastq format raw reads using the CASAVA software. The raw reads of fastq format were firstly processed through in-house Perl scripts. The clean reads were obtained by removing reads containing adapter, reads containing poly-N, and low quality reads in which a base number of Qphred less than or equal to twenty accounted for more than fifty percent of the entire read length. At the same time, Q20, Q30 and the GC content of the clean data were calculated. 

### 2.3. Read Mapping and Identification of DEGs 

The reference genome and gene model annotation files were downloaded from the genome website (ftp://ftp.citrusgreening.org/annotation/OGSv2.0/) [25]. Hisat2 v2.0.5 (https://anaconda.org/biobuilds/hisat2) was used to build the index of the reference genome and align the paired-end clean reads with the reference genome [26]. This generated a database of splice junctions based on the gene model annotation file. The expression levels of the genes were calculated by fragments per kilobase of transcript per million fragments mapped (FPKM). Differential expression analysis between *C*Las-free groups and *C*Las-infected groups were performed using the DESeq2 R package (1.16.1). DESeq2 provides statistical routines for determining differential expression according to the negative binomial distribution. The resulting *P*-values were adjusted using the Benjamini–Hochberg method. A Corrected Genes *P*-value of 0.05 and an absolute |log2 (fold change)| of 0 were set as the threshold for significantly differential expression. The hierarchical cluster analysis of differentially expressed genes (DEGs) was conducted using Genesis software (http://genome.tugraz.at/genesisclient_download.shtml).

### 2.4. Gene Ontology (GO) and KEGG Enrichment Analysis of DEGs

Gene Ontology (GO) is a tool used for gene annotation by collecting defined, structured, controlled vocabulary [27]. KEGG (Kyoto Encyclopedia of Genes and Genomes) is a database used to categorize associated gene sets into appropriate pathways [28]. The clusterProfiler R package, which implements the GO terms, was used for the enrichment analysis of DEGs, in which the gene lengths of these DEGs were corrected [29]. KEGG pathway enrichment analysis for DEGs was performed using the clusterProfiler R package [30]. A *P*-value of < 0.01 was set as the threshold. 

### 2.5. Reverse Transcription Quantitative PCR (RT-qPCR) Analysis

In order to verify the reliability of the transcriptome data, the expression levels of 14 randomly selected DEGs were examined by RT-qPCR. All of the primers used are listed in Appendix A. PCR reactions were prepared containing 10 μL of SYBR Ⅱ, 8 μL of ddH_2_O, 0.5 μL of forward primer, 0.5 μL of reverse primer and 1.0 μL of cDNA template. The reactions were performed on the LightCycle^®^96 PCR Detection System (Roche, Basel, Switzerland). The relative expression level of each gene was calculated using the 2^−∆∆Ct^ method. Three biological replicates were conducted for each sample. *D. citri glyceraldehyde-3-phosphate dehydrogenase* (*GAPDH*) was used as a reference gene. The expression patterns were compared by ANOVAs, followed by Fisher’s protected least significant difference (LSD) tests [31].

## 3. Results

### 3.1. Detection of CLas Infection in the Midgut of D. citri

*D. citri* midguts were dissected in DEPC-water using a dissecting needle. The results showed that the detached midguts were relatively intact (Figure 1A). To ensure the integrity of the midgut samples for transcriptome sequencing, genomic DNA was isolated, and PCR was performed using the OI1 primer. The primer sequences are shown in Table 1. Analysis of the PCR products on agarose gels showed that the CLas-infected midguts exhibited a clear OI1 band, of 1160 bp in length, that was not present in the control group (Figure 1B) [24]. These results indicated that the selected midgut samples could be used for further transcriptome sequencing.

### 3.2. Illumina Sequencing and Read Assembly

According to the transcriptome analysis, we generated 62,582,728, 58,399,847, 56,575,636, 57,180,938, 63,335,414 and 54,887,010 raw reads from the *C*Las-infected and uninfected groups, respectively. The raw sequencing data were deposited in NCBI SRA under the accession numbers SRX7026384, SRX7026385, SRX7026386, SRX7026387, SRX7026388 and SRX7026389. After stringent quality assessment and data filtering, 61,521,780, 57,557,156, 55,768,652, 55,707,048, 62,309,800 and 53,907,044 clean reads were obtained. The Q30 (sequencing error rate < 0.1%) and Q20 (sequencing error rate < 1%) were more than 92% and 97%, respectively. The GC contents were 40.57%, 40.36%, 40.91%, 37.74%, 39.94% and 39.39%, respectively (Table 1). Therefore, the accuracy of the sequencing data was sufficient for further analysis. 

### 3.3. Identification of DEGs in Response to CLas Infection

In total, 798 DEGs were identified in the midgut of *C*Las-infected *D. citri* compared with the uninfected *D. citri*. Among these DEGs, 499 were upregulated and 279 were downregulated (Figure 2A). Basing on the log10^(RPKM+1)^ values of the two groups, we performed hierarchical clustering of all the DEGs to determine the expression patterns of the identified genes (Figure 2B). These results indicated that *C*Las infection altered the transcriptional profiles of the DEGs. 

### 3.4. GO and KEGG Enrichment Analysis

GO and KEGG enrichment analyses were conducted to further investigate the functions of the DEGs [32]. In this study, a total of 106 upregulated DEGs and 73 downregulated DEGs were performed for GO enrichment analysis. The GO enrichment analysis revealed that a total of 18 upregulated DEGs were mainly associated with the cytoplasm, 20 upregulated DEGs were associated with the organonitrogen compound metabolic process and 14 upregulated DEGs were related to the carbohydrate metabolic process (Figure 3A, Appendix A); a total of 20 downregulated DEGs were mainly involved in transmembrane transport and 14 downregulated DEGs were associated with transmembrane transporter activity (Figure 3B, Appendix A). KEGG pathway analysis is useful for researching the complex biological functions of genes [33]. According to the KEGG pathway enrichment analysis, a total of seven, three and four upregulated DEGs were significantly enriched in ribosome, proteasome and oxidative phosphorylation, respectively (Figure 4A, Appendix A). A total of six downregulated DEGs were significantly enriched in endocytosis (Figure 4B, Appendix A).

### 3.5. Validation of DEGs at the Transcriptional Level

To validate the reliability of the transcriptome sequencing data, the relative expression levels of 14 DEGs involved in different functions were analyzed by RT-qPCR (Figure 5). These results were consistent with the transcriptome data. For example, the gene E3 ubiquitin-protein ligase (DcitrP055490.1) was upregulated in both the transcriptome data and the RT-qPCR analysis, with a similar fold change. However, 40S ribosomal protein S9 (DcitrP013835.1) and heat shock protein 70 (DcitrP017415.1) were upregulated in the midgut after *C*Las infection, while they had no obvious change in the RT-qPCR analysis (Figure 5A). The linear regression analysis of the correlation between RT-qPCR and transcriptome showed an R2 value of 0.936 and a corresponding slope of 1.0208 (Figure 5B). These results suggested a significant correlation between RT-qPCR and the transcriptome data. 

### 3.6. Analysis of DEGs Associated with Ubiquitination, the Immune Response and the Ribosome

Based on the transcriptome analysis, many DEGs associated with ubiquitination, the immune response and ribosomes were altered in the uninfected groups and the *C*Las-infected groups (Table 2, Figure 6). For the ubiquitination analysis, a total of 13 DEGs were identified. Among them, eight DEGs (61.5%) were upregulated in the midgut after *C*Las infection. In addition, five DEGs were downregulated, including ubiquitin-associated domain-containing protein 1, ubiquitin-like modifier-activating enzyme ATG7, E3 ubiquitin-protein ligase RNF8, E3 ubiquitin-protein ligase MARCH2 and ubiquitin-conjugating enzyme E2J2. For the immune response analysis, 11 DEGs were obtained after *C*Las infection in the midgut; nine genes (81.8%) were upregulated except for Toll-like 8B and Beat protein. For the ribosome analysis, a total of 15 DEGs were identified, and most of the genes were upregulated, except for 28S ribosomal protein S27. 

### 3.7. Analysis of DEGs Involved in Endocytosis, the Cytoskeleton and Insecticide Resistance

Based on the GO and KEGG enrichment analyses, the DEGs involved in endocytosis, the cytoskeleton and insecticide resistance were identified (Table 3, Figure 7). For endocytosis, six (85.7%) DEGs were downregulated, and only one gene was upregulated. For the cytoskeleton analysis, a total of 15 genes were screened, and most of the proteins were upregulated after *C*Las infection. However, tyrosine protein kinase receptor torso and dynein heavy chain were downregulated. For insecticide resistance, a total of 19 DEGs were found; of these DEGs, 13 (68.4%) were upregulated and six (31.6%) were upregulated, in the midgut after *C*Las infection.

## 4. Discussion

HLB is devastating citrus production worldwide and belongs to a phloem-limited α-proteobacterium. However, the study of this bacterium is highly difficult due to the difficulties in culturing it in vitro [34]. The management of HLB depends on the removal of infected trees and the control of the *D. citri*. Chemical insecticides are currently employed as the primary management strategy, but the widespread use of insecticides leads to serious resistance in *D. citri* [35]. The study of the interaction between *C*Las and *D. citri* is promising for the control of HLB. The transmission of *C*Las in *D. citri* is a long-term process that is composed of an acquisition access period (APP), a period of latency and an inoculation access period (IAP) [13]. During *C*Las entry into the *D. citri*, the midgut is an important immune barrier for defense against bacterial infection. In a previous report, draft genome sequencing revealed that *D. citri* contained the Toll and JAK/STAT pathways, but lacked genes for the IMD pathway response to Gram-negative bacterial infection [36]. *D. citri* transmits *C*Las, beginning from a few days to a week after acquisition, and for the lifetime of the vector, suggesting that *C*Las have developed mechanisms to avoid psyllid cellular and humoral immune defenses. In *Galleria mellonella*, *Bacillus thuringiensis* infection significantly influenced antioxidant activity and the level of lipid peroxidation in the larval midgut [37]. Therefore, we considered that the *D. citri* midgut would play critical roles in the interaction between *C*Las and *D. citri*. In this study, we performed RNA-pooling sequencing, and a total of 62,582,728, 58,399,847 and 56,575,636 raw reads were obtained from *C*Las-free groups, and 57,180,938, 63,335,414 and 54,887,010 raw reads were obtained from *C*Las-infected groups. These results showed that the number of raw reads for *C*Las-free groups and *C*Las-infected groups were similar. Previous studies revealed that pooling sequencing did not represent the population variations in gene expression levels, but it could save costs and limit starting material. However, stringent false discovery rates (FDRs) and the high-throughput validation of DEGs should be considered [38]. After assembly, a total of 778 DEGs were identified in the midgut after *C*Las infection according to comparative transcriptome analysis. Many pathogens can induce the upregulation of host gene expression. Xiong et al. identified many immunity-related genes encoding pattern recognition receptors, signal modulators and immune effectors after the injection of the fungal pathogen *Beauveria bassiana* and the Gram-negative bacterium *Enterobacter cloacae* [39]. Tang et al. performed transcriptome analysis in *Musca domestica* larvae inoculated with a mixture of *Escherichia coli* and *Staphylococcus aureus*. The results showed that many genes involved in innate immunity were induced following infection [40]. In this study, KEGG enrichment analysis showed that upregulated genes were significantly enriched in ribosomes, oxidative phosphorylation and proteasomes. Downregulated genes were significantly enriched in endocytosis. 

### 4.1. Ubiquitination, the Immune Response and Ribosomes May Play Important Roles in the Midgut Response to CLas Infection

Ubiquitination occurs through a series of reactions catalyzed by different enzymes, including Ub-activating E1, Ub-conjugating E2 and Ub-ligase E3 [41]. Ubiquitination plays an important role in the recognition and clearance of some invading bacteria. Many bacterial effectors enable them to interfere with the host’s ubiquitination system, and thus to achieve successful infection [42]. Wang et al. confirmed that ubiquitin modification had multiple effects on the host immune system against Salmonella infection [43]. In this study, a total of 13 ubiquitination-related genes were identified; among them, eight genes were upregulated and five genes were downregulated after *C*Las infection. E3 ubiquitin-protein ligase was upregulated in the *D. citri* midgut following *C*Las infection. Zhang et al. revealed that zebrafish bloodthirsty member 20 with E3 ubiquitin ligase activity was involved in the immune response against bacterial infection [44]. Therefore, we speculated that E3 ubiquitin-protein ligase might play an important role in *C*Las infection. Many ubiquitinated proteins are recognized by the proteasome and are then further degraded [45]. The ubiquitin proteasome system is a key signaling pathway in the host response to bacterial or viral infection. Isaacson et al. revealed that host cells could utilize the ubiquitin-proteasome system to counteract viral infections through the generation of target structures recognized by T cells [46]. Yu et al. also identified eight differentially expressed proteins associated with ubiquitination in the *B. mori* midgut after *B. mori* nucleopolyhedrovirus (BmNPV) infection [47]. In this study, four proteasome-related proteins were upregulated after *C*Las infection, including proteasome activator complex subunit 3-like, proteasome subunit beta type, proteasome subunit alpha type and 26S proteasome non-ATPase regulatory subunit 12. We speculated that *C*Las bacteria invading the midgut might be recognized by the *D. citri* immune system. Meanwhile, *C*Las could activate host ubiquitination to eliminate immune-related proteins. 

*C*Las, as Gram-negative bacteria, activate the host immune system after invading the midgut [48]. Thus, the midgut immune response may play an important role in the defense against pathogen infection [49]. In total, 11 genes related to immune responses were differentially expressed in the *C*Las-infected groups. Serine proteases (SPs) are involved in both the prophenoloxidase (PPO) activation cascade and the Toll immune signaling pathway, especially those with a clip domain. The proteases are secreted into the hemolymph as inactive precursors and transform into the active protein involved in the melanization reaction requiring specific proteolytic cleavage [50]. In this study, we found that three genes belonging to the serine protease family were upregulated after *C*Las infection, including CLIP domain-containing serine protease 2-like, CLIBP-serine protease 5 and CLIBP-serine protease 4, partial. In insects, PPO is activated through a serine protease cascade upon the recognition of pathogen-associated molecular patterns. In previous research, two PPOs were identified from the genome database of *D. citri*. We considered that CLIP domain-containing serine protease 2-like might regulate *D. citri* PPOs to activate melanization to defend against *C*Las infection [27]. 

In the process of host invasion by pathogens, many proteins are utilized to achieve replication. Ribosomal proteins, in conjunction with rRNA, make up the ribosomal subunits involved in the cellular process of translation [51]. In this study, a total of 15 DEGs related to ribosomes were screened, comparing the *C*Las-infected group to the *C*Las-free group. Among these genes, 14 were upregulated after *C*Las infection, except for 28S ribosomal protein S27 (mitochondrial). Yu et al. also found that some ribosome-associated proteins were upregulated after BmNPV infection [47]. These results indicate that ribosomal proteins are likely to play crucial roles in the response to *C*Las infection. 

### 4.2. Endocytosis, the Cytoskeleton and Insecticide Resistance May Play Crucial Roles in the Midgut Response to CLas Infection

Clathrin-mediated endocytosis is mainly involved in the selective and facilitated internalization of cell surface receptors [52]. Many pathogens can take advantage of the endocytosis machinery in the cytosol for replication [53,54,55]. However, the specific mechanism of endocytosis between viruses and bacteria is different. Bacteria can secrete some proteins or components that allow the modification of the internalization vacuole to permit an intravacuolar lifestyle with concomitant replication [56]. For some viruses, endocytes can help transport incoming particles deep into the cytoplasm unobstructed by cytoplasmic crowding and obstacles such as the cytoskeleton [57]. Interestingly, seven genes associated with endocytosis were found, and most of them were downregulated after *C*Las infection, except for heat shock protein 70. During *C*Lso infection of the potato psyllid, *C*Lso cells were observed between the basal lamina and the basal epithelial cell membranes [58]. For example, ADP ribosylation factor proteins comprise a group of five Ras-related GTPases that are thought to function as regulators of membrane traffic. Interestingly, *C*Las are Gram-negative bacteria, but they are not pathogenic to *D. citri*. In addition, many studies revealed that *C*Las could be detected in the hemolymph from *C*Las-infected *D. citri* [23]. Therefore, we speculated that after *C*Las invading, *D. citri* might inhibit the expression of endocytosis-related genes in the midgut to prevent the further transmission of *C*Las. Moreover, *C*Las might avoid the host immune system by endocytosis. 

The cytoskeleton is critical for the maintenance of cell shape, cell motility and intracellular transport, and bacterial and viral infections require the cytoskeleton [59]. Bacteria in the process of invasion and proliferation, at all stages of the intracellular bacterial life cycle, have the same three-dimensional cytosolic space containing the cytoskeleton [60]. In this instance, a total of 15 DEGs associated with the cytoskeleton were identified, and most were upregulated in the *C*Las-infected groups compared to the *C*Las-free groups. As a major protein constituent of cytoskeletal filaments, tubulin is involved in many vital cellular processes, including cell motility, cellular division and cytokinesis [61]. These results indicated that cytoskeleton-related genes played an important role in the process of *C*Las infection in *D. citri*. 

## 5. Conclusions

By using transcriptome sequencing, we identified 778 genes differentially expressed in the midgut between the *C*Las-infected groups and *C*Las-free groups. KEGG and GO enrichment analyses revealed that 80 DEGs were associated with ubiquitination, the immune response, ribosomes, endocytosis, the cytoskeleton and insecticide resistance. This study provides a foundation for further research to investigate the mechanisms of *C*Las invasion of the *D. citri* midgut.

## Figures and Tables

**Figure 1 insects-11-00171-f001:**
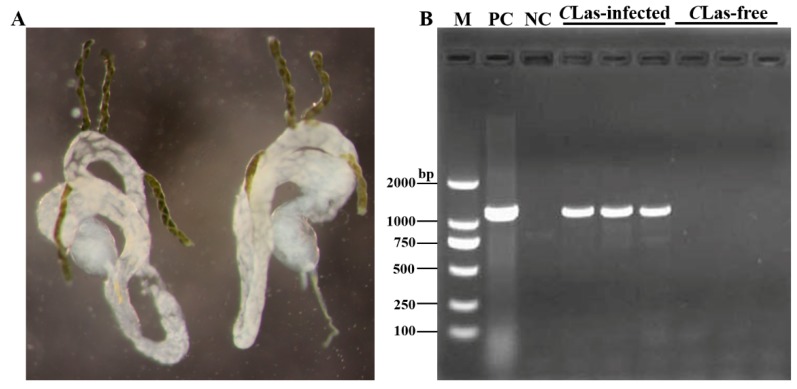
The structure of the *D. citri* midgut, and the PCR detection of *Candidatus* Liberibacter asiaticus (*C*Las). (**A**) The structure of the whole *D. citri* midgut was observed under a Lecia S8AP0 stereomicroscope. (**B**) Agarose gel electrophoresis analysis in *C*Las-free groups and *C*Las-infected groups. M, 2000 DNA marker; PC, positive control; NC, negative control.

**Figure 2 insects-11-00171-f002:**
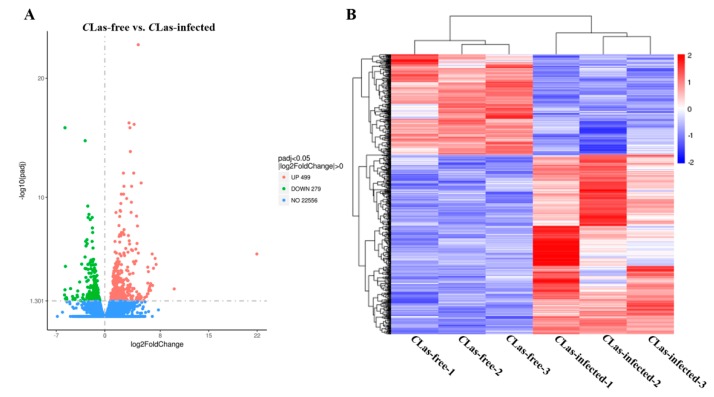
The identification and hierarchical cluster analysis of differentially expressed genes. (**A**) A scatter diagram for each gene. The blue, red and green points represent no difference in expression, upregulated genes and downregulated genes, respectively. (**B**) Hierarchical clustering of differentially expressed genes (DEGs) between the *C*Las-free groups and *C*Las-infected groups. Columns indicate different samples. Rows represent different DEGs. Blue bands indicate a low expression level, and red bands indicate a high gene expression level.

**Figure 3 insects-11-00171-f003:**
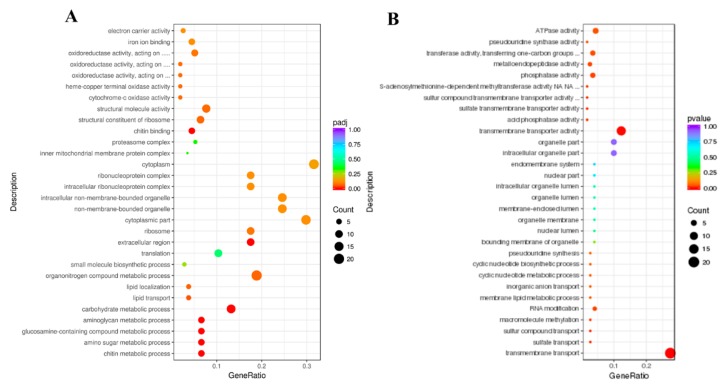
Gene Ontology (GO) enrichment analysis of DEGs. A scatter diagram of GO categories. The *X*-axis indicates the gene ratio. The *Y*-axis indicates different categories. (**A**) Upregulated DEGs. (**B**) Downregulated DEGs.

**Figure 4 insects-11-00171-f004:**
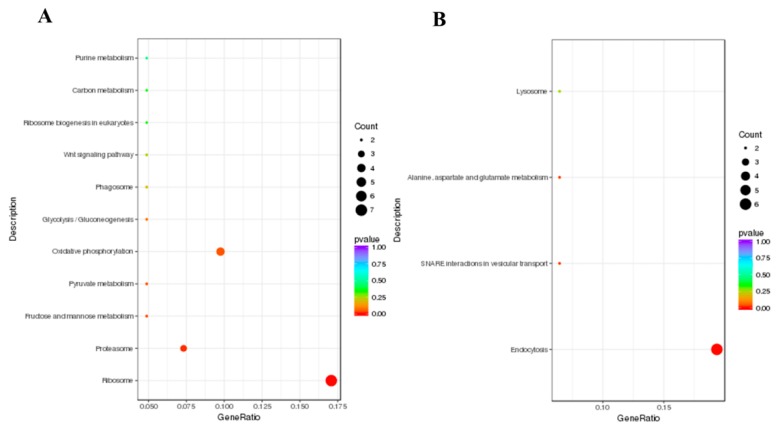
KEGG (Kyoto Encyclopedia of Genes and Genomes) enrichment analysis of DEGs. A scatter diagram of KEGG pathways. The *X*-axis indicates the gene ratio. The *Y*-axis indicates different pathways. (**A**) Upregulated DEGs. (**B**) Downregulated DEGs.

**Figure 5 insects-11-00171-f005:**
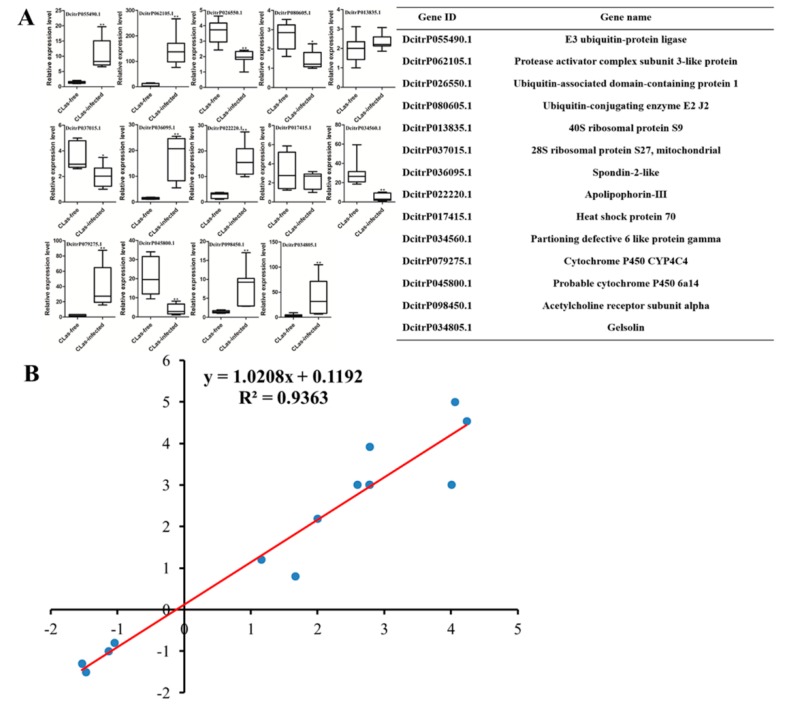
The correlation between the gene expression ratios obtained from the transcriptome data and RT-qPCR data. (**A**) The differential expression levels of 14 differentially expressed genes in the *C*Las-free and *C*Las-infected *D**. citri* midgut. The relative expression levels were calculated using the 2^−∆∆Ct^ method. Statistical analysis was performed using the SPSS software. The significant differences are indicated by * (*p* < 0.05) or ** (*p* < 0.01); (**B**) Lineage analysis between the transcriptome and RT-qPCR data. The ratios obtained by RT-qPCR (*Y*-axis) were plotted against the ratios obtained by the transcriptome (*X*-axis).

**Figure 6 insects-11-00171-f006:**
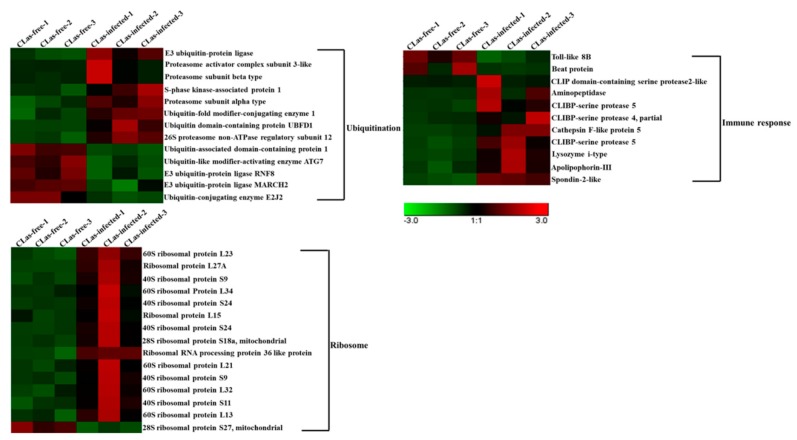
Hierarchical analysis for DEGs related to ubiquitination, the immune response and ribosomes between *C*Las-free and *C*Las-infected *D. citri* midgut. DEG expression is shown with a pseudocolor scale (from −3 to 3), with a red color indicating high expression levels and a green color indicating low expression. Each group represents three biological replicates.

**Figure 7 insects-11-00171-f007:**
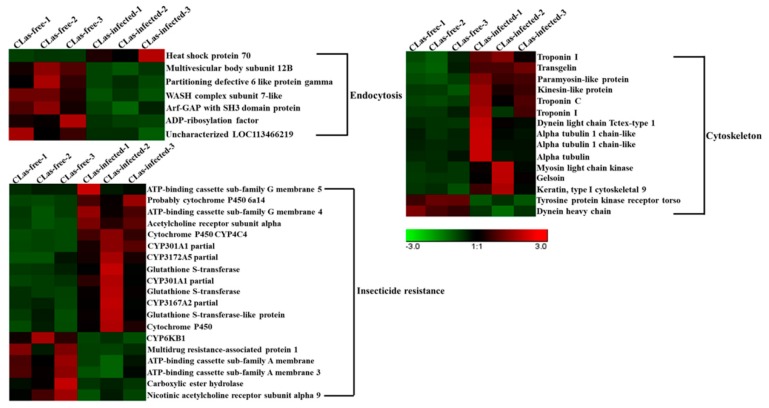
Hierarchical analysis for DEGs associated with endocytosis, the cytoskeleton and insecticide resistance between *C*Las-free and *C*Las-infected *D. citri* midgut. A hierarchical cluster analysis was performed using the Genesis software. DEG expression is shown with a pseudocolour scale (from −3 to 3), with the red colour indicating high expression levels and the green color indicating low expression. Each group represents three biological replicates.

**Table 1 insects-11-00171-t001:** A summary of the transcriptomes in the different treatments in *D. citri*.

Sample	*C*Las-Free-1	*C*Las-Free-2	*C*Las-Free-3	*C*Las-Infected-1	*C*Las-Infected-2	*C*Las-Infected-3
Total raw reads	62,582,728	58,399,847	56,575,636	57,180,938	63,335,414	54,887,010
Total clean reads	61,521,780	57,557,156	55,768,652	55,707,048	62,309,800	53,907,044
Q20	97.25%	97.14%	97.37%	97.59%	97.16%	97.48%
Q30	92.45%	92.20%	92.72%	93.13%	92.24%	92.90%
GC percent	40.57%	40.46%	40.91%	37.74%	39.94%	39.39%
Total map	43,131,403 (70.11%)	40,754,962 (70.81%)	40,200,207 (72.08%)	32,534,301 (58.4%)	41,738,828 (66.99%)	35,207,811 (65.31%)
Unique map	34,876,548 (56.69%)	32,834,545 (57.05%)	32,347,622 (58.0%)	26,881,942 (48.26%)	33,556,647 (53.85%)	28,376,185 (52.64%)
Multiple map	8,254,855 (13.42%)	7,920,417 (13.76%)	7,852,585 (14.08%)	5,652,359 (10.15%)	8,182,181 (13.13%)	6,831,626 (12.67%)
Read 1 map	17,503,161 (28.45%)	16,493,012 (28.66%)	16,205,676 (29.06%)	13,455,363 (24.15%)	16,865,739 (27.07%)	14,200,948 (26.34%)
Read 2 map	17,373,387 (28.24%)	16,341,533 (28.39%)	16,141,946 (28.94%)	13,426,579 (24.1%)	16,690,908 (26.79%)	14,175,237 (26.3%)
Positive map	17,505,192 (28.45%)	16,467,596 (28.61%)	16,228,967 (29.1%)	13,459,583 (24.16%)	16,829,779 (27.01%)	14,230,468 (26.4%)

**Table 2 insects-11-00171-t002:** Differentially expressed genes upon *C*Las infection involved in ubiquitination, the immune response and the ribosomes comparing *C*Las-infected groups and *C*Las-free groups.

Gene ID	Gene Description	*C*Las-Free FPKM	*C*Las-Infected FPKM	Log2Fold Change
Ubiquitination				
DcitrP024565.1	Proteasome subunit alpha type	8.590624	18.91098	1.142678
DcitrP057820.1	S-phase kinase-associated protein 1	5.07435	12.83314	1.338850457
DcitrP057935.1	Ubiquitin-fold modifier-conjugating enzyme 1	7.195371	14.55307	1.015273471
DcitrP026550.1	Ubiquitin-associated domain-containing protein 1	10.65557	3.702037	−1.531855619
DcitrP066965.1	Ubiquitin-like modifier-activating enzyme ATG7	29.43184	14.80442	−0.993594896
DcitrP055490.1	E3 ubiquitin-protein ligase	0.333095	2.28688	2.782229875
DcitrP062105.1	Proteasome activator complex subunit 3-like protein	2.290002	13.83469	2.595512129
DcitrP059695.1	Proteasome subunit beta type	1.753526	3.587919	2.414010482
DcitrP081240.1	Ubiquitin domain-containing protein UBFD1	8.095103	15.50153	0.93331621
DcitrP077930.1	26S proteasome non-ATPase regulatory subunit 12	11.5905	20.46284	0.818448568
DcitrP069975.1	E3 ubiquitin-protein ligase MARCH2	78.9481	38.47541	−1.034157481
DcitrP069805.1	E3 ubiquitin-protein ligase RNF8	203.473	97.33118	−1.064416343
DcitrP080605.1	Ubiquitin-conjugating enzyme E2 J2	24.049	10.95165	−1.132484873
Immune response				
DcitrP022770.1	CLIP domain-containing serine protease 2-like	0.15049	2.354921	3.964562534
DcitrP022775.1	CLIPB-serine protease 4	0.569921	5.485922	3.275711152
DcitrP046955.1	Aminopeptidase	0.223217	1.070738	2.276052354
DcitrP016435.1	Lysozyme i-type	4.803546	19.42617	2.011353707
DcitrP097315.1	CLIPB-serine protease 5	2.744837	6.396924	1.223892409
DcitrP029315.1	CLIPB-serine protease 5-RA	1.551907	3.488869	1.163879263
DcitrP022220.1	Apolipophorin-Ⅲ	82.6109	184.9353	1.162289806
DcitrP036745.1	Toll-like 8B	66.77792	39.1133	−0.771504993
DcitrP077945.1	Beat protein	1.732096	0.063739	−4.609706023
DcitrP077305.1	Cathespin F-like protein 5-RA	0.014575	0.0578843	5.280394143
DcitrP036095.1	Spondin-2-like	0.036088	0.612192	4.060891064
Ribosome				
DcitrP073005.1	60S ribosomal protein L23	28.24808	126.5357	2.162205397
DcitrP093430.1	Ribosomal protein L27A	98.15065	435.6547	2.152761801
DcitrP013835.1	40S ribosomal protein S9	4.209662	13.43044	1.666951923
DcitrP008150.1	60S ribosomal protein L34	17.65276	51.35397	1.536118663
DcitrP059870.1	40S ribosomal protein S24	155.3869	427.2229	1.458686099
DcitrP062275.1	Ribosomal protein L15	101.5892	271.7571	1.418811677
DcitrP059845.1	40S ribosomal protein S24	21.11473	55.26206	1.385121791
DcitrP068410.1	28S ribosomal protein S18a, mitochondrial	43.2703	102.078	1.235994788
DcitrP007765.1	Ribosomal RNA processing protein 36 like protein	4.525509	10.3174	1.186528712
DcitrP019375.1	60S ribosomal protein L21	206.8503	463.4805	1.16374505
DcitrP027780.1	40S ribosomal protein S9	419.7915	883.7374	1.073779521
DcitrP025515.1	60S ribosomal protein L32	533.5668	1100.274	1.044015195
DcitrP002745.1	40S ribosomal protein S11	102.8649	207.1952	1.009873361
DcitrP094130.1	60S ribosomal protein L13	48.45802	96.80903	0.997246375
DcitrP037015.1	28S ribosomal protein S27, mitochondrial	6.245021	3.043301	−1.04281238

**Table 3 insects-11-00171-t003:** Differentially expressed genes upon CLas infection involved in endocytosis, the cytoskeleton and insecticide resistance, comparing *C*Las-infected groups and *C*Las-free groups.

Gene ID	Gene Description	*C*Las-Free FPKM	*C*Las-Infected FPKM	Log2Fold Change
Endocytosis				
DcitrP017415.1	Heat shock protein 70	4.970658	34.09295	2.779406588
DcitrP097875.1	Multivesicular body subunit 12B	24.38566	8.356045	−1.541534978
DcitrP031105.1	WASH complex subunit 7-like protein	17.45894	7.028736	−1.315473724
DcitrP034560.1	Partioning defective 6 like protein gamma	26.1217	9.444002	−1.46961334
DcitrP007925.1	ADP-ribosylation factor	16.66552	6.056527	−1.459442252
DcitrP079685.1	Arf-GAP with SH3 domain	6.149781	2.706943	−1.176262825
novel.2039	Uncharacterized LOC113466219	48.38901	23.34858	−1.053542718
Cytoskeleton				
DcitrP027020.1	Paramyosin-like protein	6.933056	15.75305	1.185025088
DcitrP028800.1	Troponin I	29.9198	67.87614	1.18133546
DcitrP031990.1	Tyrosine protein kinase receptor torso	2.771104	1.295568	−1.104829478
DcitrP084355.1	Troponin I	0.433339	3.163564	2.881484544
DcitrP056675.1	Alpha tubulin 1 chain-like	9.993052	47.73973	2.256687099
DcitrP062610.1	Myosin light chain kinase	0.72003	2.930362	2.021891929
DcitrP046925.1	Transgelin	9.711448	22.08303	1.186590264
DcitrP015360.1	Kinesin-like protein	1.97132	4.45914	1.180135096
DcitrP016785.1	Alpha tubulin 1 chain-like	6.096831	53.22063	3.126504327
DcitrP034805.1	Gelsolin	17.55555	120.6549	2.780374779
DcitrP011480.1	Troponin C	0.928841	6.230001	2.755828695
DcitrP027235.1	Dynein light chain Tctex-type 1	0.84466	5.503671	2.705884305
DcitrP081065.1	Alpha tubulin 1 chain-like	1.522667	7.639856	2.330954816
DcitrP062790.1	Keratin, type I cytoskeletal 9	3.797606	10.28138	1.432892752
DcitrP002545.1	Dynein heavy chain	1.317832	0.439357	−1.557423656
Insecticide resistance				
DcitrP079275.1	Cytochrome P450 CYP4C4	0.331885	6.304564	4.238246725
DcitrP045800.1	Probable cytochrome P450 6a14	0.172821	2.734463	4.010391446
DcitrP079270.1	CYP301A1 partial	1.019659	11.5746	3.500092719
DcitrP079255.1	CYP301A1 partial	2.045632	13.91084	2.763107035
DcitrP039115.1	ATP-binding cassette sub-family G member 5	0.349581	1.914687	2.457939089
DcitrP019570.1	Glutathione S-transferase	108.29	310.5219	1.519243483
DcitrP022100.1	CYP3167A2 partial	3.10311	8.045237	1.366858366
DcitrP071995.1	CYP3172A5 partial	1.847749	4.73452	1.355174187
DcitrP019590.1	Glutathione S-transferase	14.94143	37.15758	1.311635196
DcitrP069025.1	ATP-binding cassette sub-family G member 4	3.412652	8.348752	1.294131073
DcitrP042025.1	ATP-binding cassette sub-family A member	21.79537	9.547165	−1.187067336
DcitrP090960.1	Multidrug resistance-associated protein 1	56.71604	22.4627	−1.335398633
DcitrP013660.1	CYP6KB1	17.94354	6.599974	−1.448082102
DcitrP078955.1	ATP-binding cassette sub-family A member 3	81.79196	9.226527	−1.549025159
DcitrP051650.1	Carboxylic ester hydrolase	8.538477	2.508855	−1.773370392

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
