# Peer review of "Transcriptome Analyses of Diaphorina citri Midgut Responses to Candidatus Liberibacter Asiaticus Infection"

_insects, 2020, doi:10.3390/insects11030171_

Round 1

Reviewer 1 Report

The authors performed RNA-seq of ACP and determined the differentially expressed genes in the midguts by comparing the CLas-infected and Clas-uninfected ACP. The 778 DEGs were identified and they were associated with ubiquitination, the immune response, ribosome, endocytosis, the cytoskeleton, and insecticide resistance. Although the novelty is limited, this work provides foundation data for future studies on the interaction between ACP and CLas. I have some comments as shown below.

Line 16-17. I don’t think the primary goal of this work is to determine the number of genes that respond to CLas. The authors should revise the open question of this study. Line 20, it’s functional enrichment analysis or functional annotation analysis but not functional analysis. Line 73. What are DEPs? I guess it is a mistake. Revise it. Line 85, remove this sentence. It’s unnecessary here. Line 87, add “in the midguts” before “at the mRNA level” Line 95, what does the “long time” mean? Specify it. Line 97, describe the method or the kit for DNA extraction. Line 98-102, I feel confused about the determination of the rate of CLas infection. The author should clearly describe it. I guess the authors may check a marker gene of CLas to do it, but I am not sure. So, more details were needed. Line 103, how many midguts were collected for one sample? Line 163, in this paragraph, the authors should tell us how many unigenes were identified. Figure 5, I suggest the authors make graphs with the FPKM profiles of the 14 genes and arrange these graphs with the RT-qPCR results together. In that case, we can easily compare the consistence. Line 229, the 9, but not 10 genes were upregulated. Revise it.

Author Response

  1. Line 16-17. I don’t think the primary goal of this work is to determine the number of genes that respond to C The authors should revise the open question of this study.

Reply: Thanks for reviewer’s valuable comments. We have revised the related descriptions in previous manuscript, seeing Line 19.

  1. Line 20, it’s functional enrichment analysis or functional annotation analysis but not functional analysis. 

Reply: Thanks for reviewer’s thoughtful comments. We have revised “Functional analysis” into “Functional annotation analysis” in previous manuscript, seeing Line 23.

  1. Line 73. What are DEPs? I guess it is a mistake. Revise it.

Reply: Thanks for reviewer’s thoughtful comments. We have deleted the “DEPs” in previous manuscript, seeing Line 82.

  1. Line 85, remove this sentence. It’s unnecessary here.

Reply: Thanks for reviewer’s thoughtful and valuable suggestions. We have removed this sentence, seeing Line 95 to Line 96.

  1. Line 87, add “in the midguts” before “at the mRNA level”.

Reply: Thanks for reviewer’s valuable suggestions. We have added the related content in previous manuscript, seeing Line 97.

  1. Line 95, what does the “long time” mean? Specify it.

Reply: Thanks for reviewer’s valuable suggestions. We have removed the sentence “After a long time”, seeing Line 105.

  1. Line 97, describe the method or the kit for DNA extraction. 

Reply: Thanks for reviewer’s valuable comments. In this study, the genome DNA of D. citri was extracted using TIANamp Micro DNA kit. We have added the name of kit in previous manuscript, seeing Line 108 to Ling 109.

  1. Line 98-102, I feel confused about the determination of the rate of CLas infection. The author should clearly describe it. I guess the authors may check a marker gene of CLas to do it, but I am not sure. So, more details were needed.

Reply: We are sorry for unclear descriptions in previous manuscript. We have added the detailed descriptions in previous manuscript, seeing Line 106 to Line 112. In this study, based on previous report (Ultrastructures of Candidatus Liberbacter asiaticus and its damage in huanglongbing (HLB) infected citrus), we used the HLB agent specific primer sets, OI1/OI2c to amplify the 16S ribosomal DNA fragment.

  1. Line 103, how many midguts were collected for one sample?

Reply: We are sorry for unclear descriptions in previous manuscript. We have added the relevant content in previous manuscript, seeing Line 118 to Line 119. In this study, to meet the RNA concentrations for transcriptome sequencing, a total of nine hundred D. citri were collected and divided three groups, and each group contained three hundred D. citri to obtain the midgut.

  1. Line 163, in this paragraph, the authors should tell us how many unigenes were identified.

Reply: We are sorry for the incorrect descriptions in previous manuscript. We have revised the related descriptions, seeing Line 137 to Line 151. In this study, according to transcriptome sequencing and data filtering, the clean reads were obtained. To date, the Diaphorina citri genome has been sequenced, Hisat2 v2.0.5 siftware was used to build the index of the reference genome and align the paired-end clean reads with the reference genome. Therefore, the clean reads were not assembled into the unigenes, but were directly mapped to the reference genome.

  1. Figure 5, I suggest the authors make graphs with the FPKM profiles of the 14 genes and arrange these graphs with the RT-qPCR results together. In that case, we can easily compare the consistence.

Reply: Thanks for reviewer’s thoughtful suggestions. We have added the Lineage analysis between the transcriptome and RT-qPCR data, seeing Line 259 to Line 260.

  1. Line 229, the 9, but not 10 genes were upregulated. Revise it. 

Reply: Thanks for reviewer’s thoughtful comments. We have revised the incorrect descriptions, seeing Line 277.

Reviewer 2 Report

The paper of Yu et al. describes the detection of differentially expressed genes in the midgut of Diaphorina citri, the transmission vector of the bacterium Candidatus Liberibacter spp which cause the huanglongbing disease. The paper is very interesting and well written. However, going through the text several points raised my attention.

Materials and Methods

Line 95: “after a long time” should be described in a more scientifically way.

Line101-102: After sample collection, did you store the samples in some way (directly -80 till processing) or did you directly process the samples? Is there any possible bias due to RNA degradation? The protocols used to extract the DNA and run the PCR should be briefly described or at least referenced. It is not clear to me the initial and final number of samples used to run the experiment. I discovered later that only 3 healthy samples were compared against 3 infected samples. You should stress a bit more the fact that this study should be considered just as a pivotal one.

Line 117: you mentioned a “first quality check” using an in-house perl script? Is it possible to have details about this step? Is it used to remove adapters?

Line 118: what do you mean with “low quality reads”?

Line 124-128: what is the main point of gene annotation with the NCBI nr database? Did you perform a BLAST search to gain a higher fraction of assembled sequences similar to some genes in order to retrieve more functions? In case, can you provide details? Can you describe a bit KO and GO annotations (in which way you did the analyses)? Do the mentioned software perform a search by sequence similarity? If yes, which are the resources (databases) they use? References about Nr, KO, GO Interpro and Kobas are missing. Also, the version of tools and databases are missing.

Line 130 Which tool did you use to compute FPKM?

Line 137: About GO and KEGG-based gene enrichment analysis, did you use the annotation sets retrieved as described at line 126-127? To understand the background used in the analyses (numbers of genes) and the impact of multiple corrections, how many genes did you annotate and how many annotation sets (GO, different classes; KEGG) did you retrieve from each database? (The total number of genes is UP + DOWN + NO as reported in Figure 2?; it is not stated in the text). This is just for completeness and to understand the numeric details of enrichment analyses. Did you perform any FDR or Bonferroni correction (I think you did since in Figure 3 and 4 there is a “padj”)? What was the alpha used? Can you provide details?

Line 139: did you use a specific distance measure in this clustering analysis?

Results

Line 154: OL1 primers are not reported in Table 1. Table 1 reports statistics about sequenced reads.

Line 155. I would add the reference to Figure 1B at the end of the first sentence.

Table 1: I do not understand the meaning of “positive map”.

Line 179: “More DEG were upregulated…” Can you provide numbers to justify this sentence? Is this difference significant and is there any meaning of this difference that should be discussed?

Figure 2. I would specify at line 135, and also in the figure legend, that “padj” refers to the adjusted P-value.

Figure 3 and 4: Do “counts” in the legend mean the number of input DE genes over-represented for a specific GO/KEGG?. In order to claim over-representation, do you always have at least two input genes in those over-represented gene sets? To me, it seems no (I see some dots that say 1 gene). In that case, you can not state the enrichment of some functions. For completeness, you should provide also supp. tables reporting statistics of enrichment including the input genes belonging to each over-represented gene set.

Line 214:  “Suggest a significant correlation”. You should provide a number and a P-value.

Line 222-223. You claimed that DEGs are associated with ubiquitination, immune response, ribosome. However, while ribosome and ubiquitination (proteasome?) are among the over-represented functions, the “immune system” does not appear to me as obviously present. Are some of the over-represented GO/KEGG in some way related to immune response functions? Are the genes reported in Table 2 and Figure 6 the one that are also over-represented? Otherwise, why did you chose to report those genes?

Discussion

Line 273-276. I do not understand why you are trying to explain differences in the number of sequenced reads. It should be related to the chip/machine since a fixed number of reads can be analysed. I do not see the necessity to explain this difference. Or at least, it is not obvious to me.

278-280. I do not understand this sentence “To our surprise…”.  Does it mean that usually, after bacterial infection, genes are down-regulated? Probably I miss some knowledge in the field. I suggest removing “To our surprise”.

Author Response

  1. Line 95: “after a long time” should be described in a more scientifically way.

Reply: Thanks for reviewer’s valuable suggestions. We have removed the sentence “After a long time”, seeing Line 105.

  1. Line101-102: After sample collection, did you store the samples in some way (directly -80 till processing) or did you directly process the samples? Is there any possible bias due to RNA degradation? The protocols used to extract the DNA and run the PCR should be briefly described or at least referenced. It is not clear to me the initial and final number of samples used to run the experiment. I discovered later that only 3 healthy samples were compared against 3 infected samples. You should stress a bit more the fact that this study should be considered just as a pivotal one.

Reply: Thanks for reviewer’s valuable suggestions. Because the D. citri individuals are very small in size, to meet the RNA concentrations for transcriptome sequencing, we need to dissect hundreds of D. citri. In order to avoid RNA degradation, the collected midgut samples were stored in RNA latter and froze in -80 ℃. In addition, all RNA from different samples was tested to see whether they met transcriptome sequencing. In this study, the genome DNA of D. citri was extracted by using TIANamp Micro DNA kit. Using the above genome DNA as template, PCR was performed using the HLB agent specific primer sets OI1/OI2c based on previous report (Ultrastructures of Candidatus Liberibacter asiaticus and its damage in huanglongbing (HLB) infected citrus). We have added the detailed descriptions in previous manuscript, seeing Line 108 to Line 112. For CLas-infected groups, a total of nine hundred D. citri was divided into three groups. In each group, the three hundred midgut samples were mixed for RNA extraction. The CLas-free groups were treated as the same way. Therefore, transcriptome sequencing was performed containing three biological replicates. The relevant information was added to previous manuscript, seeing Line 118 to Line 119.   

  1. Line 117: you mentioned a “first quality check” using an in-house perl script? Is it possible to have details about this step? Is it used to remove adapters?

Reply: Thanks for reviewer’s thoughtful comments. In this study, using in-house perl script, we removed reads containing adapter, reads containing poly-N and low quality reads to obtain the clean reads. In brief, the first step is to use in-house perl source code, integrate the data quality control software with the drawing script, and compile it into a computer-recognized binary. The quality control was performed using in-house perl script according to fastQC software. All input files should be presented as FastQ format. And then a directory of total quality control results will be generated, so as to visually see the problematic status of the results and quickly direct to each file, such as the number of reads, the frequency of length distribution, base mass distribution, GC content and adapter removing. The result is presented in the form of chart to visually show the accusation status of each result part. The specific FastQC based on the linked (http://wap.sciencenet.cn/blog-3406804-1161193.html?mobile=1).

  1. Line 118: what do you mean with “low quality reads”?

Reply: Thanks for reviewer’s thoughtful comments. The low quality reads indicated that the base number of Qphred less than or equal to 20 accounted for more than 50% of the entire read length.

  1. Line 124-128: what is the main point of gene annotation with the NCBI nr database? Did you perform a BLAST search to gain a higher fraction of assembled sequences similar to some genes in order to retrieve more functions? In case, can you provide details? Can you describe a bit KO and GO annotations (in which way you did the analyses)? Do the mentioned software perform a search by sequence similarity? If yes, which are the resources (databases) they use? References about Nr, KO, GO Interpro and Kobas are missing. Also, the version of tools and databases are missing.

Reply: We are sorry for the incorrect descriptions. According to other reviewer’s suggestions, we have rewritten the content of “GO and KEGG enrichment analysis” in previous manuscript, seeing Line 152 to Line 158. For functional annotation of DEGs, GO and KEGG enrichment analysis was performed by using the clusterProfiler R package.    

  1. Line 130 Which tool did you use to compute FPKM?

Reply: Thanks for reviewer’s thoughtful suggestions. In this study, the featureCounts v1.5.0-p3 was used to count the reads numbers mapped to each gene. And then FPKM of each gene was calculated based on the length of the gene and reads count mapped to this gene.

  1. Line 137: About GO and KEGG-based gene enrichment analysis, did you use the annotation sets retrieved as described at line 126-127? To understand the background used in the analyses (numbers of genes) and the impact of multiple corrections, how many genes did you annotate and how many annotation sets (GO, different classes; KEGG) did you retrieve from each database? (The total number of genes is UP + DOWN + NO as reported in Figure 2?; it is not stated in the text). This is just for completeness and to understand the numeric details of enrichment analyses. Did you perform any FDR or Bonferroni correction (I think you did since in Figure 3 and 4 there is a “padj”)? What was the alpha used? Can you provide details?

Reply: Thanks for reviewer’s thoughtful comments. According to other reviewer’s suggestions, we have rewritten the content of “GO and KEGG enrichment analysis” in previous manuscript, seeing Line 152 to Line 158. As described in Table S2, GeneRatio indicates the ratio between the number of DEGs annotated to GO terms and the total DEGs. BgRatio indicates the ratio between the number of background genes annotated to GO terms and the total background genes. For example, a total of 106 upregulated DEGs was performed for KEGG analyais. However, a total of 4957 background genes were analyzed. We have added the detailed descriptions in previous manuscript, seeing Line 229 to Line 240

  1. Line 139: did you use a specific distance measure in this clustering analysis?

Reply: Thanks for reviewer’s valuable suggestions. In this study, we used Euclidian distance in this clustering analysis. Euclidian distance is variant to both, adding and multiplying all vectors with a constant factor. It is also variant to the dimension of the vector values, for example if missing reduce the dimension of certain vectors.   

  1. Line 154: OL1 primers are not reported in Table 1. Table 1 reports statistics about sequenced reads.

Reply: Thanks for reviewer’s thoughtful suggestions. In Table 1, all primers were used for RT-qPCR analysis. We have added the sequences of OI1 primers in previous manuscript, seeing Line 111 to Line 112

  1. Line 155. I would add the reference to Figure 1B at the end of the first sentence.

Reply: Thanks for reviewer’s thoughtful suggestions. We have added the reference in previous manuscript, seeing Line 193.

  1. Table 1: I do not understand the meaning of “positive map”.

Reply: Thanks for reviewer’s valuable comments. The meaning of “positive map” indicated that the number of reads mapped to positive-sense strand of reference genome.

  1. Line 179: “More DEG were upregulated…” Can you provide numbers to justify this sentence? Is this difference significant and is there any meaning of this difference that should be discussed?

Reply: Thanks for reviewer’s thoughtful comments. According to further consideration, we have deleted this sentence “More DEG were upregulated…” in previous manuscript, seeing Line 218 to Line 219.

  1. Figure 2. I would specify at line 135, and also in the figure legend, that “padj” refers to the adjusted P-value.

Reply: Thanks for reviewer’s thoughtful and valuable comments. In this study, the P-value indicates the value of the significance test. The padj refers to the P-value after multiple hypothesis testing correction.

  1. Figure 3 and 4: Do “counts” in the legend mean the number of input DE genes over-represented for a specific GO/KEGG?. In order to claim over-representation, do you always have at least two input genes in those over-represented gene sets? To me, it seems no (I see some dots that say 1 gene). In that case, you can not state the enrichment of some functions. For completeness, you should provide also supp. tables reporting statistics of enrichment including the input genes belonging to each over-represented gene set.

Reply: Thanks for reviewer’s thoughtful and valuable comments. For GO enrichment analysis, we selected the most significant 30 terms to construct a scatter diagram. If the number of GO terms less than 30, we presented all terms. The KEGG enrichment analysis was performed according to the same way. We have added the statistics of GO/KEGG enrichment analysis including the imput genes to Table S2 and Table S3.

  1. Line 214:  “Suggest a significant correlation”. You should provide a number and a P-value.

Reply: Thanks for reviewer’s thoughtful comments. According to reviewer’s suggestions, we have performed the linear regression analysis of the correlation between RT-qPCR and transcriptome. The results showed an R2 value of 0.936 and a corresponding slope of 1.028. The detailed results have been added to previous manuscript, seeing Line 256 to Line 258.

  1. Line 222-223. You claimed that DEGs are associated with ubiquitination, immune response, ribosome. However, while ribosome and ubiquitination (proteasome?) are among the over-represented functions, the “immune system” does not appear to me as obviously present. Are some of the over-represented GO/KEGG in some way related to immune response functions? Are the genes reported in Table 2 and Figure 6 the one that are also over-represented? Otherwise, why did you chose to report those genes?

Reply: Thanks for reviewer’s valuable and thoughtful comments. The ubiquitination indicates ubiquitination-proteasome in this study. Based on GO and KEGG enrichment, the DEGs associated with ubiquitination and ribosomes. In previous report, insect immune response plays an important role in the defense against pathogen infection. CLas as gram-negative bacteria can activate the D. citri immune systems after invading the midgut (Innate immune system capabilities of the Asian citrus psyllid, Diaphorina citri). However, the D. citri genome sequencing revealed that D. citri lacked many immune-related genes and signal pathways, for example, the Imd pathway, most antimicrobial peptides,1,3-β-glucan recognition proteins(GNBPs), and complete peptidoglycan recognition proteins (Annotation of the Asian Citrus Psyllid Genome Reveals a Reduced Innate Immune System ). Therefore, we also focused on the immune-related DEGs. Interestingly, we found that two serine protease showed up-regulated after CLas infection, including CLIP domain-containing serine protease 2-like and CLIPB-serine protease 4 (Table 2). In our research group, we are studying the specific functions of two genes in the process of CLas infecting D. citri midgut. Taken together, we chose to report those genes

  1. Line 273-276. I do not understand why you are trying to explain differences in the number of sequenced reads. It should be related to the chip/machine since a fixed number of reads can be analysed. I do not see the necessity to explain this difference. Or at least, it is not obvious to me.

Reply: Thanks for reviewer’s valuable comments. We have deleted the unclear descriptions, seeing Line 325 to Line 327.

  1. 278-280. I do not understand this sentence “To our surprise…”.  Does it mean that usually, after bacterial infection, genes are down-regulated? Probably I miss some knowledge in the field. I suggest removing “To our surprise”.

Reply: Thanks for reviewer’s valuable suggestions. We have deleted the sentence “To our surprise…”, seeing Line 329 to Line 331.

Reviewer 3 Report

The paper is written relatively well, with some minor tense issue throughout that should be addressed. For example L310 'We speculated' should be in present form 'We speculate'. 

Introduction: There are two sentences that seem incomplete L46-L47 'The high quality of the...' and 'Some key genes...'. It is also a little confusing to be using ACP and D. citri interchangeably, as there are several other abbreviations used. I would choose one and stick to it throughout. 

Materials and methods: In L95 the authors need to define a 'long time'. In section 2.2, the authors should specify the sample size of each of their test groups (infected and uninfected). How many biological replicates per group?  Additionally, in L117 they refer to 'in-house perl scripts' these need to be deposited in GitHub (or a similar database) and cited in the manuscript. In section 2.3, they used Trinity to assemble, but later state that there is a reference genome available. Did they perform a de novo assembly or reference guided assembly? Or a combination? This needs to be specified.

Discussion: This is good and thorough from my perspective.

Author Response

  1. L310 'We speculated' should be in present form 'We speculate'。

Reply: Thanks for reviewer’s valuable comments. We have revised “We speculated” into “We speculate”, seeing Line 362.

  1. There are two sentences that seem incomplete L46-L47 'The high quality of the...' and 'Some key genes...'. It is also a little confusing to be using ACP and D. citri interchangeably, as there are several other abbreviations used. I would choose one and stick to it throughout. 

Reply: Thanks for reviewer’s thoughtful and valuable comments. We have deleted the unclear descriptions in previous manuscript, seeing Line 51 to Line 53. In addition, we also revised all “ACP” into “D. citri”.

  1. In L95 the authors need to define a 'long time'. In section 2.2, the authors should specify the sample size of each of their test groups (infected and uninfected). How many biological replicates per group?  Additionally, in L117 they refer to 'in-house perl scripts' these need to be deposited in GitHub (or a similar database) and cited in the manuscript. In section 2.3, they used Trinity to assemble, but later state that there is a reference genome available. Did they perform a de novo assembly or reference guided assembly? Or a combination? This needs to be specified.

Reply: We are sorry for the incorrect descriptions in previous manuscript. We have deleted the sentence “After a long time”, seeing Line 105 to Line 106. In this study, a total of three biological replicates in each group were performed for transcriptome. All clean reads were mapped to reference genome rather than Trinity assembly. We have revised the related descriptions, seeing Line 140 to Line 143.

Round 2

Reviewer 2 Report

The manuscript presented by Yu et al. presents several improvements. However, the revised version raised in my mind some other concerns:

1) The original paper did not clearly detail that this is a RNA-pool sequencing. RNA levels are subjected to intra-group abundance variations, that could lead to biases when samples are pooled. Bioinformatics tools use single replicates in order to handle and model this variability in the right way. A comparison between single sample and  RNA-pool sequencing reported that is strategy is prone to a high FPR (see pubmed: PMC4515013). Pool-seq should be avoided and bar-coding of samples should be preferred.

However, I understand that this strategy is money-saving and could be used as the first approach to test a hypothesis. I suggest to clearly discuss the pros/cons of this strategy to let the readers know about potential biases.

2)  To let the reader know, it should be better to add to the text the following points:

  1. “In order to avoid RNA degradation, the collected midgut samples were stored in RNA later and froze in -80 ℃.”
  2. “The low quality reads indicated that the base number of Qphred less than or equal to 20 accounted for more than 50% of the entire read length.”

3) The part related to the NCBI nr database and KEGG orthology disappeared. To me, it is not clear what happened. KEGG orthology switched to KEGG pathway. Moreover, regarding functional enrichment there are still doubts about the results:

  1. The database used to perform the analysis is related to the fruit fly (Drosophila melanogaster) and this is not described in the main text as well as the no. of background genes used in the analysis.
  2. The authors claim over-representation of gene functions as follows: “A P-value of < 0.01 was set as the threshold”. They ignored multiple testing correction and the requirements of at least two DEG in each over-represented KEGG/GO. Tables, images and text should be modified as a consequence. Moreover, column names of Table S2 and S3 are missing and not clear for the reader (padj = Bonferroni, or BH, or other FDR-related procedures?; what are GeneRatio, BgRatio, geneID or geneName?)

Hope these suggestions could help you to improve the manuscript, in order to make it more reproducible. 

Author Response

  1. The original paper did not clearly detail that this is a RNA-pool sequencing. RNA levels are subjected to intra-group abundance variations, that could lead to biases when samples are pooled. Bioinformatics tools use single replicates in order to handle and model this variability in the right way. A comparison between single sample and RNA-pool sequencing reported that is strategy is prone to a high FPR (see pubmed: PMC4515013). Pool-seq should be avoided and bar-coding of samples shoule be preferred. However, I understand that this strategy is money-saving and could be used as the first approach to test a hypothesis. I suggest to clearly discuss the pros/cons of this strategy to let the readers know about potential biases. 

Reply: Thanks for reviewer’s valuable comments. We have added the detailed descriptions in previous manuscript, seeing Line 331 to Line 333.

  1. To let the reader know, it should be better to add to the text the following points: 1. “In order to avoid RNA degradation, the collected midgut samples were stored in RNA later and froze in -80 ℃”. 2. “The low quality reads indicated that the base number of Qphred less than or equal to 20 accounted for more than 50% of the entire read length”.

Reply: Thanks for reviewer’s valuable suggestions. We have added the related descriptions in previous manuscript, seeing Line 119 to Line 120, Line 136 to Line 137.

  1. The database used to perform the analysis is related to the fruit fly (Drosophila melanogaster) and this is not described in the main text as well as the no. of background genes used in the analysis.

Reply: Thanks for reviewer’s valuable comments. In this study, GO and KEGG enrichment analysis were performed according to hypergeometric distribution principle using the clusterProfiler software. As shown in supplementary Figure, the differential gene set is the set of genes obtained by significant difference analysis and annotated to GO or KEGG database, while the background gene set is the set of all genes analyzed by significant difference and annotated to GO or KEGG database. The enrichment analysis results were that all the differentially expressed gene sets, up-regulated differentially expressed gene sets and down-regulated differentially expressed gene sets of each differentially expressed comparison combination were enriched. In addition, we performed transcriptome sequencing based on D. citri genome database.

  1. The authors claim over-representation of gene functions as follows: “A P-value of < 0.01 was set as the threshold”. They ignored multiple testing correction and the requirements of at least two DEG in each over-represented KEGG/GO. Tables, images and text should be modified as a consequence. Moreover, column names of Table S2 and S3 are missing and not clear for the reader (padj = Bonferroni, or BH, or other FDR-related procedures?; what are GeneRatio, BgRatio, geneID or geneName?)

Reply: Thanks for reviewer’s valuable and thoughtful comments. We have revised the Table 3 and Table 4, seeing Line 243 to Line 250. In addition, we have added the detailed column names in Table S2 and Table S3.

Round 3

Reviewer 2 Report

The authors improved the manuscript as requested.